# Factors Influencing the Rationing of Nursing Care in Selected Polish Hospitals

**DOI:** 10.3390/healthcare10112190

**Published:** 2022-10-31

**Authors:** Zuzanna Radosz-Knawa, Alicja Kamińska, Iwona Malinowska-Lipień, Tomasz Brzostek, Agnieszka Gniadek

**Affiliations:** 1Department of Medical and Environmental Nursing, Institute of Nursing and Midwifery, Faculty of Health Sciences, Jagiellonian University Medical College, 31-501 Krakow, Poland; 2Laboratory of Theory and Fundamentals of Nursing, Institute of Nursing and Midwifery, Faculty of Health Sciences, Jagiellonian University Medical College, 31-126 Krakow, Poland; 3Department of Nursing Management and Epidemiology Nursing, Institute of Nursing and Midwifery, Faculty of Health Sciences, Jagiellonian University Medical College, 31-501 Krakow, Poland

**Keywords:** BERNCA-R, rationing, Polish hospitals

## Abstract

Introduction: The rationalization of nursing care can be a direct consequence of the low employment rate or unfavorable working environment of nurses. Aim: The aim of the study was to learn about the factors influencing the rationing of nursing care. Methods: The study group consisted of 209 nurses working in internal medicine departments. The study used the method of a diagnostic survey, a survey technique with the use of research tools: the BERNCA-R questionnaire and the PES-NWI questionnaire (which includes the occupational burnout questionnaire). Results: The mean total BERNCA score for rationing nursing care was 1.94 ± 0.75 on a scale from 0 to 4. A statistically significant relationship was demonstrated between the work environment and the rationing of nursing care. The results of the BERNCA-R scale correlated statistically significantly and positively (r > 0) with two (out of three) subscales of the occupational burnout questionnaire (MBI—Maslach Burnout Inventory): emotional exhaustion and depersonalization (*p* < 0.001), and with all types of adverse events analyzed (*p* < 0.05). Conclusions: The higher the frequency of care rationing, the worse the assessment of working conditions by nurses, and, therefore, more frequent care rationing determined the more frequent occurrence of adverse events. The more frequent the care rationing, the more frequent adverse events occur.

## 1. Introduction

Nurse care rationing is defined as “holding back or failing to perform essential nursing tasks due to insufficient time or the level of professional skills of the staff” [1] (p. 227). Care rationalization can be a direct consequence of the low employment rate and unfavorable working environment of nurses [1]. This phenomenon was originally investigated by M. Schubert et al. [1,2] in the context of the Rationing of Nursing Care in Switzerland (RICH) research project. To measure it, Swiss researchers developed the Nursing Care Rationing Assessment Questionnaire (BERNCA-R) instrument, while the Polish adaptation of this scale was prepared by Uchmanowicz et al. in 2019 [3].

Nursing practice covers a wide range of daily tasks assigned to patient care both in hospitals and in other forms of inpatient and home care. With limited human resources, insufficient to provide adequate care for patients, nurses are forced to ration their care, using their own clinical assessment to set priorities that are not always appropriate to their knowledge and skills [4]. Research shows that in many hospitals, the number of nurses is not optimal to provide adequate care for patients [5,6], while the lack of nursing staff is associated with negative patient outcomes. Studies conducted in several countries (USA, UK, and Belgium) have shown a significant correlation between the working environment of nurses, their professional skills, and an increased number of adverse events or medical errors committed during a patient’s stay in hospital (treatment errors, falls, nosocomial infections, “failed rescue”, and mortality) [7,8,9,10,11,12,13]. Moreover, negative features of the work environment of nurses are significantly related to their dissatisfaction with work, occupational burnout, and injuries resulting in the course of the work of nursing staff as a result of adverse events [12,14,15]. Despite evidence that increasing the number of nurses is a cost-effective intervention [16,17], current economic policies in European countries may make it difficult to achieve this goal for economic and political reasons in individual countries of the European Union. In some countries, despite the financial resources of healthcare units, nurses do not want to work in hospitals due to poor working conditions [18]. It has also been shown that in many cases when nurses are unable to perform all nursing and caring activities, they prioritize those activities that provide the best possible care using the available resources [18]. The process of making decisions by nurses in situations of staff shortages whereby nursing and care procedures should be performed or omitted is rarely described in the literature on the subject. Therefore, in recent years, the need to examine the patient–nurse relationship has been demonstrated, paying attention to how nurses intellectually and physically organize and provide the necessary nursing care, how to ration it, and how to use potential opportunities with little human and financial outlay [6]. Over the past decade, attention has been paid to three areas related to the non-performance of activities in care: nursing neglect [19], nursing care omission [20], and implicit nursing care rationing [1]. Despite the differences in defining these concepts, they try to understand which nursing activities are partially or completely overlooked when resource or nurse shortages prevent the provision of comprehensive, necessary care. Omitted care is, paradoxically, a promising argument to force changes in the employment standards of nursing staff. However, it should be noted that there is still little evidence to date that the employment of support staff in healthcare teams has reduced the risk of loss of care [21]. Therefore, undertaking research in the field of nursing care rationing, which included the environments of various hospitals within one region of Poland, where nursing care is diversified, appears to be correct and worth assessing. In Poland, the problem of rationing nursing care still requires investigation, especially comparisons within and between institutions. Therefore, the aim of the study was to determine the scope of nursing care rationing and its relationship with the work environment and sociodemographic factors.

## 2. Materials and Methods

### 2.1. Research and Design

The study was multicenter, cross-sectional, and observational. The selection of the group for the research was deliberate. For logistical reasons, only hospitals from southern Poland were selected, as well as those where the heads of the institutions agreed to conduct the research. Nurses received a paper questionnaire. The subjects of the research were 209 nurses working in internal medicine departments of 10 hospitals in the southern region of Poland, with a total of 11 departments of this profile. The study was planned in such a way that all nurses working in the internal medicine wards during the study took part in it, according to the staffing that the hospital management decided was correct for the number of beds. The inclusion criteria for the study were met by 221 nurses, of which 209 nurses gave their voluntary consent to participate in the study. The role of the ward nurse in each ward was to explain the purpose of the research and to collect the completed questionnaires and return them to the research team.

After obtaining the consent of the Bioethics Committee (opinion no. 1072.6120.11.2019 of 31 January 2019), an application was made to the directors of the institutions for consent to conduct research in their subordinate units. Hospital Directors’ approval was obtained and the participation of nurses in the study was voluntary. The management of the surveyed hospitals did not provide data on the number of nurses employed in internal medicine departments. The research was conducted from June 2019 to January 2020. For the sake of clarity of the record, a simplified record of individual units selected for the study was adopted in this study. When describing the research material, the following entry was adopted: Hospital 1, Hospital 2, Hospital 3, Hospital 4, Hospital 5, Hospital 6, Hospital 7, Hospital 8, Hospital 9, Hospital 10, and Hospital 11.

### 2.2. Participants

The inclusion criteria for the study were seniority of nursing staff in the internal medicine ward: at least one full-time year, while the exclusion criteria were the following elements: no consent to participate in the study, work on a ward other than an internal medicine ward, work experience less than 1 year, part-time work. The first year is often an introductory period when the nursing staff do not perform all activities or perform them under supervision; therefore, the inclusion criterion was work experience of at least 1 year. Nurses were informed orally about the purpose and principles of the study and agreed to participate in the study.

### 2.3. Methods

In the present study, the dependent variable was nursing care rationing. Nursing care rationing was measured using the BERNCA-R questionnaire originally developed and validated by the Nursing Rationing in Switzerland study [1], and this work uses a tool that is culturally and linguistically adapted to Polish conditions [3]. Alpha correlations and correlations between Cronbach’s items were used to analyze the internal consistency of the Polish BERNCA-R questionnaire. The mean BERNCA-R total score was 1.9 points (SD = 0.74) on a 0–4 scale. Cronbach’s alpha for the one-dimensional scale was 0.96. The mean correlation between the items was 0.4 (range 0.1–0.84). The one-factor solution showed stable loads above 0.5 for almost all items of the Polish BERNCA-R questionnaire. The study with the use of the Polish BERNCA-R questionnaire showed that the tool is accurate and reliable in the study of care rationing in groups of Polish nurses. The BERNCA-R questionnaire contained 32 items listing the necessary nursing tasks, including daily nursing activities, emotional or psychosocial support, educational and rehabilitation care, safety conditions, and documentation [3]. Using a 5-point Likert-type scale (0 = no need, 1 = never, 2 = rarely, 3 = sometimes, 4 = often), respondents (nurses) were asked to rate how often in the last seven working days they had not been able to complete one or all of the 32 tasks due to insufficient time, staff, and/or skills. To calculate the mean level of presumed nursing rationing, the scores for each nurse were averaged across all 32 items. The total score ranged from 0 to 60 and the mean from 0 to 3.0 [3].

The Nurse’s Professional Satisfaction questionnaire was built from three standardized tools: PES-NWI (The Practice Environment Scale of the Nursing Work Index), MBI (Maslach Burnout Inventory), and one question taken from the Medical Office Survey on Patient Safety developed by AHRQ [22]. In order to determine the influence of working environment conditions on care rationing, the relationship between the variable of nursing care counseling and the characteristics of the working environment was measured by a question from the PES-NWI NWI questionnaire (question A3: How do you assess the conditions of your work in this hospital (sufficient staff and technical means, relations with co-workers, support of superiors)?). The independent variables were: occupational burnout of nursing staff, sociodemographic data, working environment conditions of nursing staff, and occurrence of adverse events.

The worksheet included five assessment aspects: employment adequacy, nurse–doctor cooperation, support received by nurses from management, nurses’ participation in hospital management, and support for the quality of nursing care. The respondents were asked to mark one answer for each statement. The nursing workload in the ward was determined based on the number of employed nurses, the number of patients in the ward, and the number of patients directly entrusted to the care of one nurse, as well as the workload and non-professional duties performed by nurses during the last shift. When analyzing job satisfaction, the prevailing working conditions in the hospital were assessed, including: a sufficient number of staff, technical measures, superiors’ support, and relations with co-workers. When assessing human resources, a summary assessment of four questions was used, constituting the PES-NWI subscale assessing the adequacy of nursing staff resources.

Following this, the subjective opinion of the surveyed nurses was taken into account: whether the nurses think there is enough staff to provide proper care, and whether the hospital has support and technical staff. Nine aspects of job satisfaction were assessed further on in the survey. They were also asked about potential intentions of leaving the job resulting from dissatisfaction and possible preferences in choosing a new job. The Maslach Burnout Inventory (MBI) questionnaire was used in the Nurse Satisfaction Survey to examine the burnout phenomenon, which consisted of three subscales: emotional exhaustion, depersonalization, and job satisfaction. The second part of the PES-NWI questionnaire assessed the quality of nursing care and the safety of care. This section consisted of seven groups of questions designated to understand the overall assessment of the quality of patient care and the occurrence of adverse events (patient received the wrong drug, at the wrong time or dose, pressure ulcers occurred after admission, patient falls with injury, care-related urinary tract infection, care-related blood infection, care-related pneumonia) and other events that concerned nurses or patients (complaints from patients or their families, verbal abuse of nurses by patients and/or their families or staff, physical insults of nurses by patients and/or their families or staff, work-related physical injuries of nurses. The third part contained information on the last shift that the nurses had to perform.

### 2.4. Statistical Analysis

The analysis of quantitative variables was performed by calculating the mean, standard deviation, median, and quartiles. Analysis of qualitative variables (performed by calculating the number and percentage of occurrences of each value). The comparison of the values of the qualitative variables in the groups was performed using the chi-square test (with Yates’s correction for 2 × 2 tables) or Fisher’s exact test where low expected frequencies appeared in the tables. The comparison of the values of quantitative variables in two groups was performed using the Mann–Whitney test. The comparison of the values of quantitative variables in three and more groups was performed using the Kruskal–Wallis test. Correlations between quantitative variables were analyzed using the Spearman correlation coefficient. A significance level of 0.05 was adopted in the analysis. The analysis was performed in the R software, version 4.0.3 (R Fundation for Statistical Computing, Vienna, Austria).

### 2.5. Ethical Procedures

The participation of nurses in the study was voluntary and anonymous. The study was conducted in accordance with the ethical standards of the Helsinki Declaration (64 WmA General Assembly, Fortaleza, Brazil, October 2013) and in accordance with Polish legal regulations. The research was approved by the Bioethics Committee of the Jagiellonian University (opinion no. 1072.6120.11.2019 of 31 January 2019) and the consent of the management of individual hospitals.

## 3. Results

### Characteristics of the Studied Group

The survey was conducted in a group of 209 people practicing the profession of nursing. In the analyzed study, the majority of respondents were women 83.7% (*N* = 175), and men constituted 15.3% (*N* = 34) of the respondents. Out of the surveyed group, as many as 62.7% (*N* = 131) of nurses took up additional employment. The youngest person participating in the study was 22 years old and the oldest one was 56 years old. The vast majority of nurses, 83.7% (*N* = 175) worked in shifts. Almost half of the nurses, 49.28% (*N* = 103), have higher education (Table 1).

The mean total BERNCA-R nursing rationing score was 1.94 ± 0.75 on a scale from 0 to 4. The median care rationing score was 1.94, as detailed in Table 2.

The obtained results concerning the rationing of nursing care in all examined medical care units are presented in Table 3. There were statistically significant differences (*p* < 0.05) between all analyzed medical care units compared with each other in terms of rationing nursing care. Nursing care rationing frequency was highest in Department 1 (SD = 0.63; Me = 2.53; Q1 = 2.02; Q3 = 2.89), and the lowest in Department 5 (SD = 0.46; Me = 1.22; Q1 = 1.06; Q3 = 1.5).

The most frequently rationed (taking into account the average score starting with the highest) were: talking to the patient or family (x_śr_ = 2.34), providing the patient with emotional or psychosocial support (x_śr_ = 2.24), getting to know the situation of individual patients and their care plans at the start of the shift (x_śr_ = 2.2), assessment of the needs of new patients (x_śr_ = 2.11), and activities related to the hygiene of the patient’s teeth (x_śr_ = 2.1). On the other hand, the least frequently rationed were: the hygienic hand washing procedure (x_śr_ = 1.52), the proper performance of the disinfection procedure (x_śr_ = 1.56), no delay in taking the necessary actions in the event of an unforeseen, acute or sudden change in the patient’s health due to long waiting for a doctor (x_śr_ = 1.59), and preparing the patient for examination or treatment (x_śr_ = 1.61).

A statistically significant relationship was demonstrated between the work environment and rationing nursing care. The higher the frequency of care rationing, the worse the assessment of working conditions by the nursing staff, *p* < 0.05 (Table 4).

Out of 209 respondents, 91 (43.54%) showed a high level of emotional exhaustion, 61 respondents (29.19%) were medium, and 57 respondents (27.27%) were low. Low level of depersonalization showed 106 out of 209 respondents (50.72%), 53 respondents (25.36%) had a medium level of depersonalization, and 50 respondents (23.92%) had a high level of depersonalization. Job satisfaction at a low level had 88 out of 209 respondents (42.11%), 68 respondents (32.54%) had an average level of professional satisfaction, and 53 respondents (25.36%) had a high level of professional satisfaction.

The results of the BERNCA-R scale correlated statistically significantly (*p* < 0.05) and positively (r > 0) with all analyzed types of adverse events, so it was shown that the more frequent the rationing of care, the more frequent the adverse events (Table 5). The most common adverse events that positively correlated with rationing nursing care were the occurrence of pressure ulcers in the patient after admission to the ward—the Spearman’s correlation coefficient was r = 0.31 and care-related blood infection was the least related (Spearman’s correlation coefficient was r = 0.154).

The frequency of care rationing was correlated with the occupational burnout of the surveyed nurses. The results of the BERNCA-R scale correlated statistically significantly (*p* < 0.001) and positively (r > 0) with two subscales of the occupational burnout questionnaire (MBI): emotional exhaustion and depersonalization. There was no statistically significant correlation between the occupational burnout subscale “job satisfaction” (*p* = 0.897). The results are shown in Table 6.

The linear regression model showed that there was one independent predictor of the BERNCA results (*p* < 0.05). In the age parameter, the regression score is −0.01, so each subsequent year of life reduces the BERNCA-R score by an average of 0.01 points. The R^2^ coefficient for this model was 5.03%, which means that 5.03% of the variability in the BERNCA-R score was explained by the variables taken into the model. The results are shown in Table 7.

## 4. Discussion

The reasons for rationing nursing care include, among others: reductions in medical staff employment, increased patient need for care due to technological advances, more treatment options, and more informed users of health services [23]. A patient who goes to a hospital has higher expectations and is aware that the institution should meet them, not only in terms of professional care, but also good services, e.g., safe hotel services. Such an approach, however, requires greater attention from specialists, and at the same time a change in the awareness of system users that not only the drug is important in the recovery process, but also other aspects that determine health [23]. The COVID-19 pandemic situation made it very measurably indicative of what the type of care should be, and that even the most modern drug will not work if there is no one to administer it and there is no place where the patient can receive this drug. Nursing care rationing occurs during the care process at the intersection of the nurse–patient relationship. A constraint can be viewed as the end product of clinical evaluation and decision-making processes when resources are not sufficient to provide all necessary nursing care to all patients. In this case, the responsible nurse has no choice but to rationalize certain aspects of the necessary care [23]. Scientific evidence indicates that rationing nursing care is strongly influenced by decisions made by nurses, using their clinical judgement and knowledge of how to allocate already limited resources [24,25]. Our own research confirmed the existence of care rationing in the examined internal medical care units. The mean score for all examined health care units was 1.94 (SD = 0.75; Me = 1.94; Q1–Q3 = 1.31–2.5). Similar results were obtained by Fabich in 2020, who studied nurses from the Subcarpathian voivodeship, where the average BERNCA-R score was 1.64 (SD = 0.88) [26]. The rationing of nursing care in the authors’ own research was not the same in all studies of 11 medical care units. The mean result of the BERNCA-R scale ranged from 1.39 in Hospital 5 to 2.43 in Hospital 1. This may be due to the fact that in Hospital 1, the nurse looked after an average of 13.11 patients (SD = 8.69) during the last shift, and in Hospital 5 she looked after more than a half fewer patients—6.31 (SD = 2.19). Griffiths et al. emphasized that the lack of the required number of nurses resulted in the lack of appropriate nursing care [27]. In a study conducted among nurses working in Croatia, it was also found that the problem of rationing care by nurses contributes to the low quality of care offered to patients [28]. According to Zhao, the problem of rationing care can be eliminated through good cooperation in the nursing team [29].

The authors’ own research showed a statistically significant relationship between the work environment and the rationing of nursing care. Nurses who assessed working environment conditions as poor more often rationed nursing care—the mean result on the BERNCA-R scale was 2.48 (SD = 0.78; Me = 2.46; Q1–Q3 = 2.25–2.96). On the other hand, nurses who assessed working environment conditions as the mean less frequently rationed care—the mean result on the BERNCA-R scale was 2.12 (SD = 0.67; Me = 2.19; Q1–Q3 = 1.5–2.62). Nurses who assessed the working environment conditions as good or excellent were the least likely to ration the nursing care (xś = 1.63; SD = 0.67; Me = 1.50; Q1–Q3 = 1.16–2.0). So far (May 2022), no studies have been published in Poland that could be compared to our own results.

Rafferty A.M. with team [30] showed that the level of rationalized care is influenced by the quality of the nursing work environment (i.e., adequate staff and resources, teamwork, leadership, and autonomy) and patient safety [30]. Studies by various investigators from multiple centers have provided evidence that a nursing shortage and a hostile working environment can lead to unfavorable outcomes for patients, in addition to higher cost of care, and increased morbidity and mortality [24,25,26,27,28,29,30,31,32,33,34]. Nurses who assessed their working environment conditions as poor more often rationed nursing care—the mean result on the BERNCA-R scale was 2.48 (SD = 0.78; Me = 2.46; Q1–Q3 = 2.25–2.96). On the other hand, nurses who assessed the conditions of the working environment as the mean less frequently rationed care—the mean result on the BERNCA-R scale was 2.12 (SD = 0.67; Me = 2.19; Q1–Q3 = 1.5–2.62). The least frequent rationing of nursing care was those who assessed the working environment conditions as good or excellent (xś = 1.63; SD = 0.67; Me = 1.50; Q1–Q3 = 1.16–2.0). So far (May 2022), no studies have been published in Poland that could be compared to our own results. This is due to the fact that the concept of “nursing care rationing” is relatively new, and the BERNCA-R tool was validated to Polish conditions only 3 years ago. Nevertheless, similar studies have been carried out in Europe for many years. A study by the Andersson team in Sweden found that some reasons for rationing nursing care were beyond the nurses’ control. These included, among others, an insufficient number of qualified nurses on duty and a high workload of nurses. Nursing staff working time schedules were inadequately fine-tuned, resulting in a stressful working day. Stress at work, in turn, caused the nurses to be tired and forget to perform certain tasks [35]. Nursing staff believe that it is their responsibility to maintain patient stability in a complex clinical environment with limited human resources [36]. Ignoring the needs of staff and their individual skills hinders the work of the nursing team and has a negative impact on patient care [37]. Schubert and her team point to the problem that low human resources are one of the main causes of low patient outcomes in terms of treatment and care [23].

In the present study, it was also shown that the BERNCA-R scores correlated significantly (*p* < 0.05) and positively (r > 0) with all analyzed types of adverse events, so the more frequent rationing of care, the more frequently such adverse events occurred. The most common adverse event that correlated with the rationing of nursing care was the occurrence of pressure ulcers after admission to the ward—Spearman’s correlation coefficient was r = 0.31, and the least was a care-related blood infection (Spearman’s correlation coefficient was r = 0.154). The studies by Schubert et al. obtained results similar to those obtained in their own studies, i.e., the most common adverse event resulting from rationing nursing care was the occurrence of pressure ulcers after admission to the ward [38].

In available scientific publications, authors have attempted to establish the relationship between burnout and rationing nursing care [39,40]. The phenomenon of occupational burnout can be observed in the area of southern Poland. Symptoms of burnout are in the form of emotional exhaustion, depersonalization, and low satisfaction with one’s own achievements. Nurses’ working conditions are important factors in the context of professional burnout of nurses [41]. The research conducted by Blackman et al. shows that there is a significant relationship between rationing nursing care and satisfaction with nursing work and the risk of occupational burnout [42]. The Fabich research proved that job satisfaction has a positive effect on reducing the level of nursing care rationing [26]. On the other hand, Rizo-Baeza and the team found a strong correlation between satisfaction with the work of nurses and patient satisfaction with the healthcare received [43]. Studies by Kalisch et al. proved that lower job satisfaction increased the lack of nursing care [44]. Other studies have shown that high levels of nursing care rationing were closely correlated with high levels of occupational burnout [42]. Occupational burnout contributes to the rationing of nursing care [45].

Based on this study, it was found that the more often nurses rationalized care, the greater the risk of burnout. There was no statistically significant correlation between the “job satisfaction” subscale and care rationing (*p* = 0.897). Uchmanowicz et al., who studied nurses working in cardiology departments, drew similar conclusions: the level of nursing care rationing increased with emotional exhaustion, depersonalization, job dissatisfaction, and multi-professional activity. The same research confirmed that the professional activity of nurses in several healthcare institutions affects the rationing of care, which, however, was not confirmed in the authors’ own research [46].

Multiple linear regression showed that the independent predictors of the BERNCA-R score in this study were: emotional exhaustion, depersonalization, job dissatisfaction, and multi-occupational activity (*p* < 0.001) [46]. Fabich [26] showed that a high level of nursing care rationing was closely correlated with a high level of occupational burnout [26].

Organizational factors, including those related to financial savings and leadership, have the consequences of a lack, or rationing, of nursing care. Research by Witczak et al. provided evidence that an increased risk of rationing care is associated with a deteriorated working environment for nurses [47]. This has been confirmed in our own research.

When there is not enough time, nurses are forced to prioritize care. This creates dilemmas on how to prioritize due to heavy workloads, inadequate staffing levels, unexpected events, and conflicting demands from managers.

### Limitations and Strengths of the Study

The main advantages of this study include nursing staff from several medical care units in southern Poland. The sample size is not large, but all nurses who met the inclusion and exclusion criteria participated in the study. Initially, test requests were sent to 24 units from southern Poland. The consent to conduct the research was obtained from 10 hospitals (11 departments) and the research was carried out in all these units. This research on nursing rationing in Poland is one of the first conducted, as the researchers’ tool for researching the phenomenon of nursing care rationing has only recently been validated. The statements about burnout and the lack of nursing care cannot be easily generalized as research to date covers a variety of nursing healthcare settings and very specific medical environments. In addition, the concept of health care rationing is a relatively new concept, and a tool to study this phenomenon has only recently been validated in Poland [3].

The study also has its limitations, such that only consenting entities were included in the study. Another limitation is the fact that it is limited to one type of department (internal medicine department). Additionally, for logistical reasons, only hospitals located in southern Poland were selected for research. It is planned to expand the research in the course of scientific work to include departments with different specializations and from other regions of Poland. Despite these limitations, this study provided important conclusions and could be the starting point for a broader study of nursing rationing and the importance of the nursing work environment in nursing rationing.

## 5. Conclusions

Care rationing occurs in all studied internal medicine departments; however, the level of this phenomenon varies significantly between the analyzed units. Care rationalization is correlated with the occurrence of adverse events among patients, with a lower assessment of working conditions in the ward, and, moreover, with a higher risk of emotional exhaustion and depersonalization among nursing staff. In order to better understand the phenomenon, the research should be continued in departments with different specialties and extended to a larger area. However, the conducted research indicates that the phenomenon of nursing care rationing occurs. The management of the nursing teams should take steps to reduce the side effects of emotional exhaustion and depersonalization among the nursing staff.

## Figures and Tables

**Table 1 healthcare-10-02190-t001:** Descriptive statistics/demography.

Parameter	Gruop	*N*	%
Gender	Woman	175	83.7
	Man	34	15.3
Taking up additional work	yes	131	62.7
	no	78	37.3
Shift work	yes	175	83.7
	no	34	15.3
Was the last on-call shift single?	yes	11	5.26
	no	197	94.26
Education	secondary	56	26.79
	higher	103	49.28
	specialization	50	23.92

**Table 2 healthcare-10-02190-t002:** Mean total BERNCA score.

BERNCA-R [Points]								
Point Range	*N*	Mean	SD	Median	Min	Max	Q1	Q3
0–4	209	1.94	0.75	1.94	0	4	1.31	2.5

*N*—number of nurses, SD—standard deviation, Min—minimum value, Max—maximum value, Q1—lower quartile, Q3—upper quartile.

**Table 3 healthcare-10-02190-t003:** Nursing care rationing—comparison of all surveyed medical care units.

Department	BERNCA-R (Points)			*p*
	śr ± SD	Median	Quartiles	
Department 1 (*N* = 18)	2.43 ± 0.63	2.53	2.02–2.89	*p* < 0.001
Department 2 (*N* = 20)	2.17 ± 0.82	2.22	1.69–2.67	
Department 3 (*N* = 12)	2.27 ± 0.87	2.44	1.65–3.01	
Department 4 (*N* = 25)	2.35 ± 0.64	2.44	1.94–2.69	
Department 5 (*N* = 29)	1.39 ± 0.46	1.22	1.06–1.5	
Department 6 (*N* = 24)	2.28 ± 0.46	2.2	2.02–2.48	
Department 7 (*N* = 21)	1.63 ± 0.7	1.44	1.16–2.34	
Department 8 (*N* = 16)	1.47 ± 0.54	1.47	1.31–1.7	
Department 9 (*N* = 20)	1.68 ± 0.8	1.42	1.17–2.31	
Department 10 (*N* = 11)	1.62 ± 0.45	1.56	1.44–1.88	
Department 11 (*N* = 13)	2.08 ± 0.82	2.09	1.44–2.81	

*p*—Kruskal–Wallis test between all analyzed medical care units.

**Table 4 healthcare-10-02190-t004:** Assessment of working conditions in the opinion of nurses and rationing of nursing care in the opinion of nurses.

BERNCA-R (Points)	Assessment of Working Environment Conditions (PES-NWI)			*p*
	Poor (*N* = 26)—A	Average (*N* = 85)—B	Good, Great (*N* = 97)—C	
x_ś_ ± SD	2.48 ± 0.78	2.12 ± 0.67	1.63 ± 0.67	*p* < 0.05
Me	2.46	2.19	1.50	
Q1–Q3	2.25–2.96	1.5–2.62	1.16–2.00	A > B > C

Legend: *p*—Kruskal–Wallis test x_śr_ ± SD—mean ± standard deviation, Me—median, Q1–Q3—lower quartile–upper quartile; *N*—number of nurses.

**Table 5 healthcare-10-02190-t005:** Adverse events and care rationing.

Adverse Event	BERNCA-R
	Spearman’s Correlation Coefficient
Patient received the wrong drug, at the wrong time or dose	r = 0.306, *p* < 0.001 *
Pressure ulcers occurred after admission	r = 0.31, *p* < 0.001 *
Patient falls with an injury	r = 0.199, *p* = 0.004 *
Care-related urinary tract infection	r = 0.249, *p* < 0.001 *
Care-related blood infection	r = 0.154, *p* = 0.026 *
Care-related pneumonia	r = 0.272, *p* < 0.001 *

Legend: * statistically significant relationship (*p* < 0.05).

**Table 6 healthcare-10-02190-t006:** Burnout and care rationing.

MBI	BERNCA-R
	Spearman’s Correlation Coefficient
Emotional exhaustion	r = 0.343, *p* < 0.001 *
Depersonalization	r = 0.394, *p* < 0.001 *
Occupational satisfaction	r = −0.009, *p* = 0.897

Legend: * statistically significant relationship (*p* < 0.05).

**Table 7 healthcare-10-02190-t007:** BERNCA-R and sociodemographic variables—multifactorial analysis.

Variable		Regression Parameter	95%CI		*p*
Age	(years)	−0.01	−0.019	0	0.042 *
Level of education	Medical secondary school	ref.			
	Bachelor/Master of science in nursing	0.06	−0.218	0.338	0.671
	Specialization in nursing	0.169	−0.124	0.462	0.26
Second place of job	No	ref.			
	Yes	−0.184	−0.395	0.026	0.087
Place of living	Country	ref.			
	City	0.066	−0.136	0.268	0.524

Legend: *p*—multivariate linear regression; * statistically significant relationship (*p* <0.05).

## Data Availability

The data presented in this study are available on request from the corresponding author.

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
