# Peer review of "Factors Influencing the Rationing of Nursing Care in Selected Polish Hospitals"

_healthcare, 2022, doi:10.3390/healthcare10112190_

Round 1

Reviewer 1 Report

Thank you for your submission. This is an important topic.  The following concerns are raised :

1. Line 75. What do you mean by “seems right”   Not clear

2. line 80. Sample was deliberately chosen?  What made it deliberate? Why not convenience or random sampling?

3. how were data collected?  Online? Paper and pencil?

4 what was response rate? Any incentives to participate?

5 the statistics/analyses are low level.  Was a statistician used?  Why not regression analyses since you have an outcome variable?

6  under limitations the advantage stated is a large sample. You do not have a large sample. 

Author Response

Dear Reviewer,

The authors would like to thank the reviewer for his/her thorough review of the manuscript. We believe that this revised version, which includes reviewers’ suggestions, is more accurate and communicates better the main message of the article.

We will gratefully respond to any further comments on the text.

The paragraph is marked in yellow in the manuscript.

Thank you again for taking the time to review our paper and for your constructive comments.

Best regards,

Authors

Following comments have been modified in a new version of the manuscript:

  1. Line 75. What do you mean by “seems right”   Not clear

Changed to "appears to be correct" (lines: 77).

  1. line 80. Sample was deliberately chosen?  What made it deliberate? Why not convenience or random sampling?

The study was multicentre, cross-sectional and observational. The selection of the group for the research was deliberate. For logistical reasons, only hospitals from southern Poland were selected, as well as those where the heads of the institutions agreed to conduct the research. (lines: 84-87).

The choice was based on a subjective choice of the region of Poland, and the selection of the respondents to the sample was based on the consent of hospital directors to conduct the research, moreover, it was found that they would help to obtain the most comprehensive and complete information.

  1. how were data collected?  Online? Paper and pencil?

Nurses received a paper questionnaire. (lines: 87).

4 what was response rate? Any incentives to participate?

The part of  “Materials and Methods “ were improved in accordance with the comments of the reviewer.

The ward nurse of each ward was obliged to supervise, so that every nurse who met the criteria participated in the study, which aims to highlight the problem of rationing care. (lines:92-94).

5 the statistics/analyses are low level.  Was a statistician used?  Why not regression analyses since you have an outcome variable?

The comments were complied with, table number 7 has been added with regression analysis:

The linear regression model showed that there was one independent predictors of the BERNCA results (p < .05). In the age parameter, the regression score is -0.01, so each subsequent year of life reduces the BERNCA-R score by an average of 0.01 points. The R² coefficient for this model was 5.03%, which means that 5.03% of the variability of the BERNCA-R score was explained by the variables taken into the model. The results are shown in Table VII. (lines:262-267).

6  under limitations the advantage stated is a large sample. You do not have a large sample. 

The part of  “Limitations and strengths of the study“ were improved in accordance with the comments of the reviewer.

The sample size is not large, but all nurses who met the inclusion and exclusion criteria participated in the study. Initially, test requests were sent to 24 units from southern Poland. The consent to conduct the research was obtained from 10 hospitals (11 departments) and the research was carried out everywhere. (lines:388-391).

Reviewer 2 Report

The manuscript focused an important issue in healthcare and investigated influncing factors of the rationing of nursing care. The topic is interesting,but the research design and manuscript writing need to be improved.

Abstract

What is "the beginning of the shift." ?

The relation between PES-NWI questionnaire and the occupational burnout questionnaire should be clearly stated.

What is MBI?

The abstract should be rewritten to better summarize the main points of the full text.

Introduction

The necessity of the research should be clearly stated and highlighted in this part.

The sentence “ Over the past decade have highlighted three areas of non-performance in  care: nursing neglect, nursing care omission and implicit nursing care rationing” in line 66 and 67 seemed inappropriate in grammar with unclear meaning.

Materials and Methods

Research design may need to be reconsidered. Why and how the hospitals were selected? Need hospitals in different regions be compared?Why only nursing staff in the internal medicine ward were included? The comparison with nursing staff in other wards may be meaningful and interesting? Why staff with work experience less than 1 year were excluded? The work experience of nursing staff may be an important factor of  the rationing of nursing care?

What is “safety conditions and documentation” in line 116 refered to?

Results

The data in talbe I seemed inconsistent with those in related text and some data may be wrongly stated. 

11 departments in 10 hospitals were mentioned in line 82, but hospital 11 was mentioned in table III in the results. Is this appropriate?

These had raised the reviewer concerns about overall data accuracy and consistency. 

Discussion

This section should be rearranged to discuss the innovations and key findings of the research and its theoretical and practical value in a hierarchical manner.

The novelty of the work should be discussed thoroughly through comparing with the published similar studies such as, Uchmanowicz I , Karniej P , Lisiak M , et al. The relationship between burnout, job satisfaction and the rationing of nursing care—A cross‐sectional study. Journal of Nursing Management, 2020, 28(8):2185-2195,  Uchmanowicz I, Kubielas G, Serzysko B, et al. Rationing of Nursing Care and Professional Burnout Among Nurses Working in Cardiovascular Settings[J]. Front Psychology, 2021, 10,6;12:726318,  and Witczak I, Rypicz Ł, Karniej P, Młynarska A, Kubielas G, Uchmanowicz I. Rationing of Nursing Care and Patient Safety. Front Psychol. 2021 Sep 9;12:676970. doi: 10.3389/fpsyg.2021.676970. PMID: 34566757; PMCID: PMC8458807.

Conclusion

This section should be rewritten to better summarize the main points of the full text, and propose some application directions of the research results and prospects for further research.

Author Response

Dear Reviewer,

The authors would like to thank the reviewer for his/her thorough review of the manuscript. We believe that this revised version, which includes reviewers’ suggestions, is more accurate and communicates better the main message of the article.

We will gratefully respond to any further comments on the text. The paragraph is marked in red in the manuscript. Thank you again for taking the time to review our paper and for your constructive comments.

Best regards,

Authors

Following comments have been modified in a new version of the manuscript:

Abstract

  1. What is "the beginning of the shift." ?

The comments were complied with.

 The relation between PES-NWI questionnaire and the occupational burnout questionnaire should be clearly stated.

The comments were complied with. (lines:20).

  1. What is MBI?

The comments were complied with  (lines:24-25).

 The abstract should be rewritten to better summarize the main points of the full text

The comments were complied with  (lines:15-29).

Introduction

  1. The necessity of the research should be clearly stated and highlighted in this part

The comments were complied with  (lines:67-69).

“In Poland, the problem of rationing nursing care still requires deepening, especially within and inter-institutional comparisons. Therefore, the aim of the study was to determine the scope of nursing care rationing and its relationship with work environment and sociodemographic factors”.

  1. The sentence “ Over the past decade have highlighted three areas of non-performance in  care: nursing neglect, nursing care omission and implicit nursing care rationing” in line 66 and 67 seemed inappropriate in grammar with unclear meaning.

The comments were complied with  (lines:67-69).

Materials and Methods

Research design may need to be reconsidered.

  1. Why and how the hospitals were selected?

The selection of the group was deliberate, an important determinant was the consent of the hospital director to conduct the research.

  1. Need hospitals in different regions be compared?

The comparison of hospitals from different regions is very important, because in Poland there is a division of hospitals into 3 levels of referentiality, that is: The first reference level covers hospitals providing health services in four basic medical specialties: internal medicine, general surgery, obstetrics and gynecology, paediatrics, as well as anesthesiology and intensive care. The second reference level covers provincial hospitals providing health services in four basic specialties: internal medicine, general surgery, obstetrics and gynecology, paediatrics, as well as anesthesiology and intensive care, and at least four of the following specialties: cardiology, neurology, dermatology, pathology of pregnancy and newborns, ophthalmology, ENT, trauma surgery, urology, neurosurgery, pediatric surgery and oncological surgery. The third reference level covers clinical hospitals of state medical universities or a state university that conducts teaching and research activities in the field of medical sciences, as well as research and development units subordinate to the Minister of Health and Social Welfare.

It is planned to expand the research in the course of scientific work to include departments with a different specialization and from other regions of Poland.

  1. Why only nursing staff in the internal medicine ward were included? The comparison with nursing staff in other wards may be meaningful and interesting?

The selection of departments for the research was deliberate, because they wanted to check the rationing of nursing care in internal medicine departments, which mainly accommodate elderly and mostly dependent people. In departments with such a profile, it seems extremely important to assess the phenomenon of postponement or failure to perform the necessary nursing tasks due to insufficient time or the level of professional skills of the staff.

It is planned to expand the research in the course of scientific work to include departments with a different specialization and from other regions of Poland.

  1. Why staff with work experience less than 1 year were excluded? The work experience of nursing staff may be an important factor of  the rationing of nursing care?

It was found that people working for less than a year are people who further deepen their theoretical knowledge and improve their practical skills in the field of providing health services, in particular nursing, preventive, diagnostic, therapeutic and rehabilitation services as well as health promotion, in stationary and outpatient conditions. and home health care. In the case of non-performed activities / omitted activities, in the case of a group working <one year, it may be the lack of developed dexterity and manual speed, and not the actual lack of time or staff or the occupancy of the ward with patients requiring care.

  1. What is “safety conditions and documentation” in line 116 refered to

The comments were complied with  (lines:128).

Results

  1. The data in talbe I seemed inconsistent with those in related text and some data may be wrongly stated. 

The comments were complied with . The error has been corrected.

  1. 11 departments in 10 hospitals were mentioned in line 82, but hospital 11 was mentioned in table III in the results. Is this appropriate?

The comments were complied with . The error has been corrected

  1. These had raised the reviewer concerns about overall data accuracy and consistency. 

The comments were complied with . The error has been corrected

Discussion

This section should be rearranged to discuss the innovations and key findings of the research and its theoretical and practical value in a hierarchical manner.

The novelty of the work should be discussed thoroughly through comparing with the published similar studies such as,

  • Uchmanowicz I , Karniej P , Lisiak M , et al. The relationship between burnout, job satisfaction and the rationing of nursing care—A cross‐sectional study. Journal of Nursing Management, 2020, 28(8):2185-2195
  • Uchmanowicz I, Kubielas G, Serzysko B, et al. Rationing of Nursing Care and Professional Burnout Among Nurses Working in Cardiovascular Settings[J]. Front Psychology, 2021, 10,6;12:726318, 
  • Witczak I, Rypicz Ł, Karniej P, Młynarska A, Kubielas G, Uchmanowicz I. Rationing of Nursing Care and Patient Safety. Front Psychol. 2021 Sep 9;12:676970. doi: 10.3389/fpsyg.2021.676970. PMID: 34566757; PMCID: PMC8458807.

The discussion was enriched with the research results of the indicated publications.

Conclusion

  1. This section should be rewritten to better summarize the main points of the full text, and propose some application directions of the research results and prospects for further research.

The "Conclusions” part was corrected taking into account comments. (lines:411-416).

Round 2

Reviewer 1 Report

This revision was difficult to understand. Please submit a TABLE that has the reviewers' comments and your response.  Some of the insertions are not clear... like supervisors making sure surveys were filled out?  Isn't this coercion? 

Author Response

Dear Reviewer,
The authors would like to thank the reviewer for his/her thorough review of the manuscript. We 
believe that this revised version, which includes reviewers’ suggestions, is more accurate and 
communicates better the main message of the article. 
We will gratefully respond to any further comments on the text.
The paragraph is marked in yellow in the manuscript.
Thank you again for taking the time to review our paper and for your constructive comments.
Best regards,
Authors

This revision was difficult to understand. Please submit a TABLE that has the reviewers' comments and your response. 

Previous reviewers’ comments: the statistics/analyses are low level. Was a statistician used? Why not regression analyses since you have an outcome variable?

The comments were complied with, table number 7 has been added with regression analysis:

The linear regression model showed that there was one independent predictors of the BERNCA results (p < .05). In the age parameter, the regression score is -0.01, so each subsequent year of life reduces the BERNCA-R score by an average of 0.01 points. The R² coefficient for this model was 5.03%, which means that 5.03% of the variability of the BERNCA-R score was explained by the variables taken into the model. The results are shown in Table VII. (lines:262-267).

Tab. VII BERNCA-R and sociodemographic variables—multifactorial analysis

Variable

Regression parameter

95%CI

p

Age

[years]

-0,01

-0,019

0

0,042 *

Level of education

Medical secondary school

ref.

Bachelor/Master of science in nursing

0,06

-0,218

0,338

0,671

Specialization in nursing

0,169

-0,124

0,462

0,26

Second place of job

No

ref.

Yes

-0,184

-0,395

0,026

0,087

Place of living

Country

ref.

City

0,066

-0,136

0,268

0,524

Legend: p - multivariate linear regression

* statistically significant relationship (p <0.05)

  1. Some of the insertions are not clear... like supervisors making sure surveys were filled out?  Isn't this coercion? 

LINE 91-94

The inclusion criteria for the study were met by 221 nurses, of which 209 nurses gave their voluntary consent to participate in the study. The role of the ward nurse in each ward was to explain the purpose of the research and to collect the completed questionnaires and return them to the research team.